# Clinically Preferred Videolaryngoscopes in Airway Management: An Updated Systematic Review

**DOI:** 10.3390/healthcare11172383

**Published:** 2023-08-24

**Authors:** Vikram Nedunchezhian, Ishvar Nedunchezhian, André Van Zundert

**Affiliations:** 1Department of Anaesthesia and Perioperative Medicine, Royal Brisbane and Women’s Hospital, The University of Queensland, Brisbane, QLD 4029, Australia; vikram.nedunchezhian@student.bond.edu.au; 2School of Medicine and Dentistry, Griffith University, Gold Coast, QLD 4215, Australia; ishvar.nedunchezhian@health.qld.gov.au

**Keywords:** videolaryngoscope, laryngoscope, intubation, airway management, difficult airway, critical care

## Abstract

Videolaryngoscopes (VLs) have emerged as a safety net offering several advantages over direct laryngoscopy (DL). The aim of this study is to expand on our previous study conducted in 2016, to deduce which VL is most preferred by clinicians and to highlight any changes that may have occurred over the past 7 years. An extensive systematic literature review was performed on Medline, Embase, Web of Science, and Cochrane Central Database of Controlled Studies for articles published between September 2016 and January 2023. This review highlighted similar results to our study in 2016, with the CMAC being the most preferred for non-channelled laryngoscopes, closely followed by the GlideScope. For channelled videolaryngoscopes, the Pentax AWS was the most clinically preferred. This review also highlighted that there are minimal studies that compare the most-used VLs, and thus we suggest that future studies directly compare the most-used and -preferred VLs as well as the specific nature of blades to attain more useful results.

## 1. Introduction

Airway management is one of the most critical tasks an anaesthetist will encounter in their practice. Facing an unanticipated difficult airway is a complex and stressful task for any anaesthetist and requires great skill to overcome [1]. The primary goal of Endotracheal Intubation (ETI) is to establish a patent and safe airway whilst avoiding complications such as dental injuries or trauma to the anatomy surrounding the trachea. Over the years, there has been an evolution in the shape, size and material of laryngoscopes to optimise the effectiveness and, therefore, the safety of ETI [2]. Currently, direct laryngoscopy (DL) is standard practice for ETI, despite not always yielding a positive outcome [3]. A negative outcome can be referred to as a ‘can’t intubate, can’t ventilate” scenario, potentially requiring front-of-neck access, possibly leading to complications such as hypoxia and neurological injury [4]. Although significant advancements in DL have taken place over the past few decades, predicting a difficult airway remains a challenging task for anaesthetists. 

Despite extensive pre-anaesthetic assessments and precautions being undertaken, no single parameter can accurately predict an unanticipated difficult airway [5]. Hence, it is vital that clinicians are always prepared to encounter an unanticipated difficult airway, with the literature suggesting that approximately 90% of difficult intubations are unanticipated by clinicians [6]. 

Therefore, in problematic situations, a safety net for anaesthesiologists is essential to provide the utmost care for the patient. Contemporarily, videolaryngoscopes (VLs) have provided a safeguard for anaesthetists due to the advantages they provide over DLs. Videolaryngoscopy (VL) is a modern technique that uses a camera and a light source mounted on a laryngoscope blade to enable a magnified and improved view of the larynx and glottis on a screen. With a growing body of evidence and significant technological advances, VLs have recently proven to be a popular choice of device for ETI by clinicians. A meta-analysis published in 2021 concluded that there are several advantages highlighted in using a VL over a DL [3]: Higher first-pass success: VLs increased the chance of first-pass success (successful intubation on first attempt) when compared to DLs.Fewer failed intubations: VLs resulted in fewer failed intubations when compared to DLs, even in anticipated difficult airways.Fewer hypoxemic events: VLs showed a reduction in hypoxemic events when compared to DLs.Increased glottic views: certain VLs provided an improved visualisation of the glottis in accordance with the Cormack–Lehane grade, which can reduce the likelihood of adverse events such as failed intubation or airway trauma.Less sore throat: there was a lower incidence of patient-reported sore throat post-intubation using VLs, hence possibly reducing patient anxiety with intubation.

With an increase in VL use over the past decade, it is important to juxtapose the safety and efficacy of the different types of VLs. The ideal VL should deliver a high chance of first-pass successful intubation with good glottic views in a short amount of time and be readily available and cost-effective. Our previous study published in 2016 identified that the most clinically preferred VLs were the Pentax-AWS VL for channelled VLs and the Karl Storz C-MAC for non-channelled VLs [7]. 

The aim of this paper is to expand on our previous study to establish which VL is most preferred by clinicians, to further aid decision-making for safer and more effective ETI, and to ascertain whether any changes have occurred over the past 7 years in clinician preference. 

## 2. Materials and Methods

Eligibility and Search Strategy: An extensive systematic literature review was performed on Medline, Embase, Web of Science, and Cochrane Central Database of Controlled Studies for articles published between September 2016 and January 2023. The search terms included, but were not limited to: laryngoscope, laryngoscop*, videolaryngoscope, video-laryngoscope, “video laryngoscope”, videolaryngoscop*, video-laryngoscop*, “video laryngoscop*”, glidescope, macintosh, cmac, mcgrath, airtaq, “king vision”, “pentax aws”. Results were further refined to cohort studies and randomised controlled trials. A manual review of references was also performed to ensure an elaborate search. We restricted our search to English-only articles. A PRISMA checklist was used to ensure an optimal strategy, with a summary provided in Figure 1. This study was registered in PROSPERO.

Study Selection and Data Extraction: We included studies that compared either of the following between two or more videolaryngoscopes: (a) improved glottis view (either Cormack–Lehane grade or percentage of glottic opening); (b) time to successful intubation; (c) first-pass intubation success rate; (d) use of corrective manoeuvres or adjuncts; e) final outcomes judged by authors as “preferred” and/or “best” VL. Based on the factors ‘a–e’, we determined which VL was clinically preferred for each study. Data were manually extracted and cross-reviewed by the authors to optimise interrater reliability. These data were further collated to evaluate the favoured percentage of each VL. The studies chosen were grouped into either clinical patient studies or simulation studies. Table 1 demonstrates the VLs identified through our comprehensive study selection and data extraction phase.

## 3. Results

Our search yielded 3841 studies (Medline—794 results, Embase—792 results, Cochrane Library—1543 results, Web of Science—712 results). After the removal of duplicates, studies that had undergone erratum, trials registered but not yet published and meeting abstracts not yet published, 1405 articles remained. Publications that contained unrelated context or did not meet the inclusion criteria were excluded from this study.

This review included 81 studies with a total of 50 surgical studies conducted on 6274 patients, and 31 simulation studies conducted with 1353 participants. Table 2 highlights the key elements of each surgical study, while Table 3 highlights the key elements of the simulation studies. Importantly, we identified the clinically preferred VL and blade type in each of these study groups, as demonstrated in the respective tables. 

In studies evaluating surgical patients, the CMAC was the most-used VL, with 25 out of 50 studies investigating it, closely followed by the McGrath VL (19 out of 50 studies). Of note, in surgical patients, the CMAC Macintosh blade was used 14 times, the CMAC angulated D blade was used 8 times, and the CMAC Miller blade was used 3 times. In the studies that investigated the McGrath VL, 13 used the Macintosh-style blade and 6 used the angulated blade. For studies evaluating manikins, the GlideScope was the most commonly used VL, with 14 out of 31 studies investigating it, all of which used an acute-angulated blade.

Table 4 summarises the review, with less commonly used VLs excluded. The CMAC was found to be the most-preferred non-channelled VL overall (preferred in 70% of studies that investigated CMAC), closely followed by the GlideScope (preferred in 67% of studies that investigated GlideScope). It is interesting to note that the most frequently used non-channelled VLs all scored > 50% preference in surgical studies, yet only the CMAC and GlideScope scored > 50% in simulation studies. 

For channelled laryngoscopes, the AWS was the most-preferred overall, with 69% of studies that evaluated this VL preferring it; however, only 50% of surgical studies preferred this VL.

When analysing the data, it was established that the most-used VLs were not often directly compared. Out of the 50 studies reviewing clinical scenarios, the CMAC and GlideScope were only directly compared three times, and when reviewing simulation studies, they were only compared two times. Out of the clinical studies, the CMAC was preferred in two out of the three studies, whilst in the simulation studies the CMAC was preferred in one study, and the other found the CMAC and GlideScope to be similar. When comparing the CMAC to Pentax AWS, no clinical study directly compared the two VLs, and the one simulation study that compared them showed a preference for the Pentax AWS. Figure 2 demonstrates the most commonly preferred VLs.

## 4. Discussion

This review aimed to establish which VL is most clinically preferred, with the goal of expanding on our previous review published in 2016. Similar to previously, the Pentax AWS was the most preferred overall for channelled VLs, and the CMAC was the most preferred overall for non-channelled VLs. 

However, our review discovered that the more recent literature suggests that other VLs are also gaining popularity among clinicians. For instance, our previous review suggested that the GlideScope acute-angled blade was preferred in only 41% of clinical studies and 7% of simulation studies, whereas our updated review highlights that it was preferred in 69% of clinical studies and 64% of simulation studies. Likewise, the efficacy of the McGrath VL appears to have increased drastically from the previous review, increasing from a preference rate of 25% to 63% in clinical studies, and 17% to 44% in simulation studies. We hypothesise that this increase is due to the ever-expanding use, availability and familiarity with VLs. Further to this, McGrath Blades have released Macintosh-style blades, which clinicians have historically been more familiar with. With regards to our study, it is noted that a McGrath Macintosh-style blade was used in 13 of 19 clinical studies and 7 of 9 simulation studies. Again, this change may be a reason for the growing popularity and preference for the McGrath VL. Similarly, GlideScope has also released Macintosh-style blades; however, the studies in this review all appear to use the acute-angle blade. Ultimately, clinicians are more likely to perform better with and prefer a VL that they are familiar with and use more often. 

The literature suggests that acute-angle blades should be reserved for predicted or known difficult airway situations, especially in patients with an anterior larynx [89]. Thus, the use of acute-angle VLs may be detrimental, in comparison to standard Macintosh-style blades, for the intubation of normal airways. One such reason is that acute-angle VLs only provide an indirect view and present with a sharp angle, resulting in the ETT needing to be introduced with a device such as a stylet to ensure it is able to be manipulated around the steep angle [89]. Thus, one limitation of the papers studied in this review is the comparison of acute-angle blades to Macintosh-style blades, as the clinical indication for each is different.

In addition to this, there were a minimal number of studies that directly compared the most-preferred VLs. For instance, the two most-preferred non-channelled VLs, the CMAC and GlideScope, were only directly compared three times in clinical studies and two times in simulation studies. Similarly, when attempting to discuss the CMAC and Pentax AWS, no clinical study directly compared these two VLs and only one simulation study directly compared them. This limits the generalisability of the current literature, as a direct comparison and evaluation of the most-preferred VLs are not able to be conducted based on the current literature. One potential reason hypothesised is the cost involved with comparing the more expensive and most-preferred VLs. Further to this, the financial implications of acquiring, using, and maintaining VLs may limit certain departments’ ability to use VLs, and we suggest they check what is available and suitable for their needs. We also propose that the current literature favours the most commonly used VLs, and true clinician preference would be better ascertained in future studies evaluating more head-to-head comparisons and thus, more direct parameters for clinician preference. 

Furthermore, as discussed in our previous review, it must be acknowledged that clinical studies conducted on patients will differ rather significantly compared to simulation studies performed on manikins. This is highlighted in the stark difference in the performance of some VLs in clinical studies versus simulation studies. If we base our results purely on clinical studies, it would highlight that the King Vision non-channelled and Airtraq VLs may be the most suitable to use. However, both these VLs performed quite poorly in manikin models and had a <35% preference in their respective studies. Furthermore, with a minimal direct comparison between these VLs and the aforementioned most-preferred VLs, this would not be generalisable. The CMAC, GlideScope and Pentax AWS were the only VLs preferred in ≥50% of both clinical and simulation studies. 

For future use and teaching prospects, it is important to recognise extra components that may render particular VLs more preferable. For instance, the Macintosh-style blades used in the CMAC VLs can be used as both a direct and indirect laryngoscope as the video monitor is mounted separately to the laryngoscope. Therefore, in a scenario where a clinician wants to teach with this VL, the trainee can use a direct view for education, whilst the teacher can observe through the monitor to ensure the adequate placement of the endotracheal tube. Similarly, if a difficulty is encountered, a supervisor can take over, or the anaesthesia assistant can anticipate what to do next, such as preparing a bougie. Anecdotally, this method of laryngoscopy ensures safer management and education. Similarly, the McGrath VL can serve the same teaching process; however, a key difference is that the monitor is attached to the top of the VL itself, rather than mounted separately. Furthermore, different manufacturers offer different specifications in their VLs. Some provide the option for multiple types of blades or attachments such as fibreoptic scopes to be used with one VL handle, others offer longer battery life, whereas some offer the option for single-use vs reusable blades. These additional features make particular VLs more appealing, depending on the clinicians’ preferences and requirements.

A limitation to this article is that VLs were purely chosen based on their functionality in intubating a patient or manikin, without considering the resources/equipment used. For future studies, it is important to compare the cost burden for hospitals alongside clinician preference to ensure a more complete approach towards selecting the most ideal VL. Studies in the future should also be more transparent with the specific details of the VLs being used, including the type of blade, size, reusability, and/or single-use nature of the VLs. We also advocate for future studies to more thoroughly identify patient and institutional factors that may lead to the use of, or preference for, a particular VL, with the aim to provide transparency to readers regarding scenarios where particular VLs may be more favourable. 

Furthermore, as aforementioned, a limitation in the current literature is the lack of comparison between the most-used and -preferred laryngoscopes. In addition to this, studies in the future should compare VLs with Macintosh-style blades to other VLs with Macintosh-style blades, and likewise acute-angle blades with acute-angle blades. This will allow for more streamlined and consistent results. Essentially, we advocate for standardisation criteria to ensure the appropriate comparison of devices. The studies reviewed also did not consistently describe the experience level of the operators performing intubation, and suggestions for future studies would be to highlight this to ensure transparency, as it is well-known that clinicians will prefer and perform better with equipment that they are more familiar with. To achieve optimal clinical outcomes, we advocate that clinicians choose the VL that they are most comfortable with depending on the clinical situation, as each VL offers its own set of advantages and disadvantages. We believe that VLs should become the gold standard for ETI, with several advantages highlighted over DLs. Furthermore, VLs can also allow us to transition from blind insertion techniques for procedures including temperature probes and nasogastric tubes to ‘vision-guided’ insertion, ultimately limiting the potential for wrong space insertion, with future studies potentially exploring this field.

## 5. Conclusions

The purpose of this review is to critically assess and compare the effectiveness of VLs in modern airway management, both in real patients and simulation scenarios involving manikins. This study was performed as an extension of our 2016 analysis, to deduce whether there were any changes in clinician preference over the past 6 years. This review highlighted similar results to our study in 2016, with the CMAC still being the most preferred for non-channelled laryngoscopes, closely followed by the GlideScope, and the Pentax AWS being the most preferred for channelled laryngoscopes based on the current literature. We hope this audit increases the awareness of both individual practitioners and departments of anaesthesia, to highlight the importance of VL use and to lay a platform for future studies to expand knowledge in this field.

## Figures and Tables

**Figure 1 healthcare-11-02383-f001:**
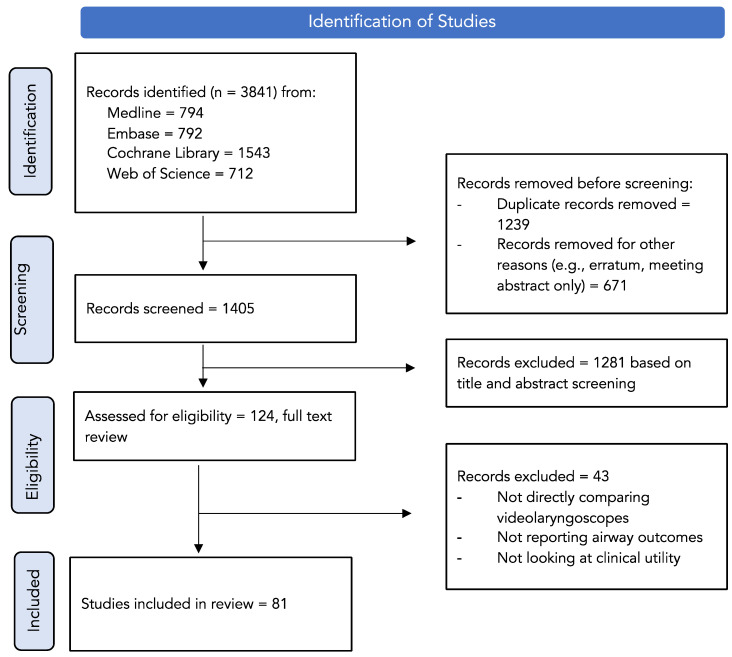
PRISMA identification, screening and selection of articles.

**Figure 2 healthcare-11-02383-f002:**
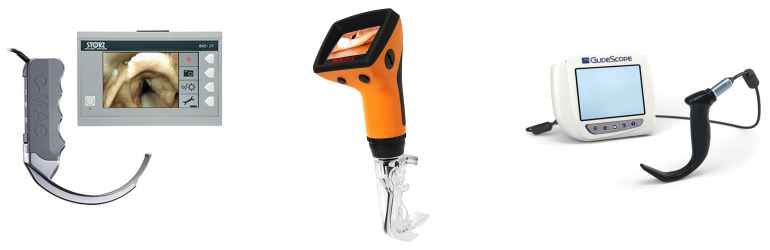
Images of the most-preferred VLs. (**Left** to **Right**): CMAC (D-Blade attachment), Pentax AWS and GlideScope.

**Table 1 healthcare-11-02383-t001:** A range of videolaryngoscopes used for airway management.

Manufacturer	Address	Videolaryngoscope	Blade Type
AirAngelBlade.org, AirAngel Project Online 3D-Printed Videolaryngoscopes	USA	AirAngel blade 3D-printed	Non-channelled, Acute-angled
Airlangga university	Surbaya, Indonesia	Wycope	Non-channelled, Acute-angled
Dahlhausen	Köln, Germany	Dahlhausen	Non-channelled, Macintosh
Industry Design Registration of Indonesia: HKI. KI.05.01.02.P00202101656 and A00202100589	Indonesia	O-Mac	Non-channelled, Macintosh
Intersurgical	Berkshire, United Kingdom	I-view	Non-Channelled, Macintosh
Karl Storz	Tuttlingen, Germany	DCI; C-MAC; C-MAC D-blade; C-MAC Pocket Monitor; CMAC Miller blade	Non-channelled, Macintosh, Miller and Acute-angled
King Systems Ambu, A/S	Ballerup, Denmark	King Vision VL channelled; non-channelled; aBlade channelled; aBLade non-channelled; Paediatric aBlade	Channelled/Non-channelled, Acute-angled
Medtronic Operational Headquarters	Minneaopolis, USA	McGrath Mac, McGrath Mac X-Blade	Non-channelled, Macintosh and acute-angulated
Pentax-AWS, Hoya Corporation	Tokyo, Japan	Pentax Airway Scope	Channelled
Prodol Meditec S.A.	Vizcaya, Spain	Airtraq; Airtraq double lumen	Channelled
Salter Labs	California, USA	Intubrite	Non-channelled, Macintosh
Smart Trach Medicare	Lucknow, India	Smart Trach	Non-channelled, Acute-angled
Soma Technology International	Bloomfield, USA	McGRATH Series 5	Non-channelled, Acute-angled
Truphatek International ltd.	Netanya, Israel	TruView; Truview EVO2; Truview PCD	Non-channelled, Acute-angled
Tuoren Medical	Henan, China	TUORen	Non-channelled, Macintosh
Venner Medical	Kiel, Germany	Venner APA channelled; non-channelled	Channelled/Non-channelled, Macintosh and Acute-angled
Verathon Medical	Bothell, WA, USA	GlideScope; GlideScope Cobalt; Glidescope Advanced Video Laryngoscope	Non-channelled, Acute-angled
Vivid Medical	Palo Alto, CA, USA	VividTrac VT-A100	Channelled

**Table 2 healthcare-11-02383-t002:** Outcomes of comparative studies in clinical patients using different types of videolaryngoscopes.

Year	First Author	No. of Patients	Setting	Intubator	Number and Types of Videolaryngoscopes Used	Outcomes Based On: A: Improved Glottis View B: Faster Time to Successful Intubation C: Higher First Attempt Intubation Success D: Less Use of Manoeuvres or Adjuncts E: Other	Clinically Preferred Videolaryngoscope	Blade Type of Most-Preferred VL
2016	Al-Ghamdi [8]	86	Surgical	Anaesthetists	3: GS, AT, KV-C	A: Equal B: GS > KV-C > AT C: Equal D: Equal E: Less sore throat in AT and KV-C than GS	GS for primary outcomes	Non-channelled, Acute-angled
2016	Alvis [9]	64	Surgical	Anaesthetic Staff	2: McGrath Mac, KV-C	A: Equal B: McGrath Mac C: McGrath Mac D: Equal	McGrath Mac	Non-channelled, Macintosh
2016	Wan [10]	90	Double Lumen Tube Surgical	Anaesthetists	2: McGrath Series 5 (Acute), AT-DL	A: Equal B: AT C: Equal D: Equal	AT	Channelled
2017	Ahmed [11]	60	Surgical Difficult patients in Neutral Position	Anaesthetists	2: CMAC, AT	A: Equal B: CMAC C: Equal D: Equal E: Less haemodynamic changes in CMAC	CMAC	Non-channelled, Macintosh
2017	Belze [12]	72	Double Lumen Tube Surgical with difficult airway	Anaesthetists	2: GS, AT	A: Equal B: Equal C: Equal D: Equal	GS = AT	GS = Non-channelled, Acute-angled AT = Channelled
2017	Kleine-Brueggeney [13]	480	Surgical	Anaesthetists	3: KV-NC, AT, APA	A: KV-NC > AT > APA B: AT > KV-NC > APA C: KV-NC > AT > APA D: -	KV-NC	Non-channelled, Acute-angled
2017	Lee [14]	140	Surgical	Anaesthetists	2: McGrath Mac, AWS	A: AWS B: Equal C: Equal D: Equal	AWS	Channelled
2017	Raza [15]	60	Surgical	Anaesthetists	2: AT, McGrath	A: Equal B: AT C: Equal D: -	AT	Channelled
2017	Sato Boku [16]	60	Surgical Nasotracheal Intubation	Anaesthetists	2: McGrath Mac, AWS	A: Equal B: McGrath Mac C: Equal D: Equal	McGrath Mac	Non-channelled, Macintosh
2017	Shravanalakshmi [17]	135	Surgical with Cervical Spine Immobilisation	Anaesthetists	3: CMAC, CMAC-D, KV-NC	A: KV-NC > CMAC-D B: CMAC > CMAC-D C: Equal D: Equal E: Ease of Laryngoscope insertion CMAC > KV-NC > CMAC-D	CMAC	Non-channelled, Macintosh
2017	Singh [18]	150	Paediatric Surgical	Anaesthetists	2: CMAC, TV-PCD	A: TV-PCD B: CMAC C: Equal D: TV-PCD	TV-PCD	Non-channelled, Acute-angled
2017	Tseng [19]	105	Surgical Nasotracheal Intubation	Anaesthetists	2: GS, AWS	A: Equal B: Equal C: Equal D: Equal	GS = AWS	GS = Non-channelled, Acute-angled AWS = Channelled
2017	Vadi [20]	93	Paediatric Surgical with Cervical Spine Immobilisation	Anaesthetic Trainees	2: GS-Cobalt, DCI	A: Equal B: Equal C: Equal D: -	GS-Cobalt = DCI	GS = Non-channelled, Acute-angled DCI = Non-channelled, Macintosh
2017	Vargas [21]	42	Surgical Difficult Airway	Anaesthetists	2: Imago-V-C, GS	A: Equal B: Equal C: Equal D: Imago-V E: Less force required with Imago-V-C	Imago-V-C	Channelled
2018	Abdelgalel [22]	120	Intensive Care Unit	ICU Physicians	2: GS, AT	A: Equal B: Equal C: Equal D: -	GS = AT	GS = Non-channelled, Acute-angled AT = Channelled
2018	Ajimi [23]	60	Double Lumen Tube Surgical	Anaesthetists	2: AT, AWS	A: Equal B: AT C: Equal D: -	AT	Channelled
2018	Cavus [24]	168	Pre-hospital	Emergency Physicians	3: CMAC-PM, KV-C, APA	A: Equal B: Equal C: CMAC-PM + APA > KV-C D: - E: Handling concerns with KV-C	CMAC-PM + APA	CMAC-PM = Non-channelled, Acute-angled APA = Non-channelled, Macintosh
2018	Chanchayanon [25]	40	Surgical	Anaesthetic Residents	2: GS, McGrath Series 5 (Acute)	A: Equal B: GS C: Equal D:-	GS	Non-channelled, Acute-angled
2018	El-Tahan [26]	133	Double Lumen Tube Surgical	Anaesthetists	3: GS, AT, KV-C	A: Equal B: AT > GS C: Equal D: AT > GS	AT	Channelled
2018	Gupta [27]	80	Surgical Neonates and Infants	Anaesthetists	2: CMAC-Miller, TV-PCD	A: Equal B: CMAC-Miller C: CMAC-Miller D: AT > GS	CMAC-Miller	Non-channelled, Miller
2018	Mendonca [28]	200	Surgical Neutral and ‘Sniffing’	Anaesthetists	2: KV-C, CMAC-D	A: KV-C > CMAC-D in Neutral B: Equal C: Equal D: - E: Modified Difficult Intubation Score Equal	KV-C	Channelled
2018	Mishra [29]	80	Nasotracheal Intubation Surgical	Anaesthetists	2: KV-NC, TV-PCD	A: Equal B: Equal C: Equal D: Equal	KV-NC = TV-PCD	KV-NC = Non-channelled, Acute-angled TV-PCD = Non-channelled, Acute-angled
2018	Yoo [30]	106	Paediatric Nasotracheal Intubation	Anaesthetists	2: AWS, McGrath Mac	A: Equal B: Equal C: Equal D: Equal	AWS = McGrath Mac	AWS = Channelled McGrath Mac = Non-channelled Macintosh
2019	Akbas [31]	80	Morbidly Obese Surgical	Anaesthetists	2: McGrath Mac, CMAC	A: Equal B: CMAC C: Equal D: Equal E: Better Haemodynamic Response in CMAC	CMAC	Non-channelled, Macintosh
2019	Blajic [32]	180	Obstetric Caesareans	Anaesthetists	2: CMAC, KV-C	A: KV-C B: Equal C: Equal D: CMAC E: Easier to use	CMAC	Non-channelled, Macintosh
2019	Chae [33]	123	Nasotracheal Intubation Surgical	Anaesthetists	2: AWS, McGrath Mac	A: McGrath Mac B: Equal C: Equal D: Equal	McGrath Mac	Non-channelled, Macintosh
2019	Markham [34]	225	Anticipated Difficult Airway Surgical	Anaesthetic Residents	3: GS-AVL, KV-C, KV-NC	A: Equal B: Equal C: GS-AVL + KV-NC > KV-C D: Equal	GS-AVL + KV-NC	GS-AVL = Non-channelled, Acute-angled KV-NC = Non-channelled, Acute-angled
2019	Roh [35]	120	Nasotracheal Intubation with Manual In-line Stabilisation Surgical	Anaesthetists	2: AWS, McGrath Mac	A: Equal B: McGrath Mac C: Equal D: Equal E: Less Bleeding McGrath Mac	McGrath Mac	Non-channelled, Macintosh
2019	Sahajanandan [36]	63	Anticipated Difficult Airway in Obese Patients Surgical	Anaesthetists	2: KV-C, CMAC-D	A: Equal B: Equal C: CMAC-D D: -	CMAC-D	Non-channelled, Acute-angled
2019	Suzuki [37]	287	Emergency Intubation Emergency Department and Intensive Care Unit	Emergency Physicians, Intensive Care Physicians, Anaesthetists and Residents	3: KV-C, AWS, McGrath Mac	A: - B: Equal C: AWS + McGrath Mac > KV-C D: -	AWS + McGrath Mac	AWS = Channelled McGrath Mac = Non-channelled, Macintosh
2019	Zhu [38]	94	Nasotracheal Intubation Anticipated Difficult Airway	Anaesthetists	2: KV-NC, McGrath Mac (acute)	A: Equal B: Equal C: Equal D: Equal	KV-NC = McGrath Mac	Non-channelled, Acute-angled McGrath Mac = Non-channelled, Macintosh
2020	Brozek [39]	110	Obese Patients Surgical	Anaesthetists	2: KV-C, GS	A: Equal B: Equal C: GS D: Equal	GS	Non-channelled, Acute-angled
2020	Huang [40]	89	Double Lumen Tube Surgical	Anaesthetists	2: GS, CMAC-D	A: CMAC-D B: CMAC-D C: Equal D: CMAC-D	CMAC-D	Non-channelled, Acute-angled
2020	Kaur [41]	120	Surgical	Anaesthetists	2: McGrath Mac, TV	A: Equal B: Equal C: Equal D: Equal	McGrath Mac = TV	McGrath Mac = Non-channelled, Macintosh TV = Non-channelled, Acute-angled
2020	Pappu [42]	120	Surgical Difficult Airway	Anaesthetists	2: TV-EVO2, CMAC-D	A: Equal B: CMAC-D C: - D: CMAC-D	CMAC-D	Non-channelled, Acute-angled
2020	Sen [43]	60	Surgical with Cervical Spine Immobilisation	Anaesthetists	2: TV, KV-NC	A: Equal B: KV-NC C: Equal D:- E: Easier Intubation with KV-NC	KV-NC	Non-channelled, Acute-angled
2021	Chandrashekaraiah [44]	60	Surgical	Anaesthetists	2: GS, CMAC-D	A: Equal B: GS C: Equal D: -	GS	Non-channelled, Acute-angled
2021	Chandy [45]	100	Surgical with Cervical Spine Immobilisation	Anaesthetists	2: KV-C, CMAC-D	A: - B: Equal C: Equal D: KV-C E: KV-C Easier	KV-C	Channelled
2021	Gupta [46]	140	Neonates and Infants Surgical	Anaesthetists	2: CMAC-Miller, McGrath Mac	A: CMAC-Miller B: Equal C: Equal D: -	CMAC-Miller	Non-channelled, Miller
2021	Mani [47]	116	Surgical with Manual In-line Stabilisation	Anaesthetists	2: CMAC-D, AT	A: Equal B: Equal C: Equal D: CMAC-D	CMAC-D	Non-channelled, Acute-angled
2021	Sepmiko [48]	270	Surgical	Anaesthetists	2: O-Mac, McGrath Mac	A: Equal B: O-Mac C: O-Mac D: O-Mac	O-Mac	Non-channelled, Macintosh
2021	Sultana [49]	120	Surgical in Lateral Position	Anaesthetists	2: CMAC, AT	A: Equal B: CMAC C: Equal D: Equal	CMAC	Non-channelled, Macintosh
2021	Teo [50]	65	Surgical	Anaesthetists	2: CMAC, GS	A: Equal B: CMAC > GS C: Equal D: Equal	CMAC	Non-channelled, Macintosh
2022	Gupta [51]	60	Surgical	Anaesthetists (COVID PPE)	2: CMAC, McGrath Mac	A: Equal B: Equal C: Equal D: Equal	CMAC = McGrath Mac	CMAC = Non-channelled, Macintosh McGrath Mac = Non-channelled, Macintosh
2022	Haldar [52]	375	Surgical	Anaesthetists	2: CMAC, ST	A: - B: CMAC > ST C: Equal D: Equal E: CMAC > ST Lifting force	CMAC	Non-channelled, Macintosh
2022	Jayadi [53]	63	Surgical	Anaesthetic residents	2: CMAC, Wycope	A: - B: Equal C: - D: Equal	CMAC = Wycope	CMAC = Non-channelled, Macintosh Wycope = Non-channelled, Acute-angled
2022	Karadag [54]	100	Surgical	Anaesthetists	2: McGrath (acute), CMAC	A: Equal B: Equal C: Equal D: Equal	McGrath = CMAC	CMAC = Non-channelled, Macintosh McGrath Mac = Non-channelled, Acute-angled
2022	Kumar [55]	140	Surgical	Anaesthetists	2: McGrath Mac, KV-C	A: Equal B: McGrath Mac > KV C: Equal D: Equal	McGrath Mac	Non-channelled, Macintosh
2022	Suryatheja [56]	160	Surgical	Anaesthetic Residents	2: CMAC Miller blade size 1, CMAC Macintosh Blade size 2	A: Equal B: Equal C: Equal D: CMAC Mac > CMAC Mil	CMAC Mac	Non-channelled, Macintosh
2022	Zhang [57]	210	Surgical	Anaesthetists	2: McGrath Mac-X, CMAC	A: McGrath Mac-X > CMAC B: CMAC > McGrath Mac-X C: Equal D: Equal	McGrath Mac-X = CMAC	McGrath Mac-X = Non-Channelled, Acute-angled CMAC = Non-channelled, Macintosh

APA = Venner AP advance; AT = Airtraq; AT-DL = Airtraq double lumen; AWS = Pentax Airway Scope; CMAC = CMAC; CMAC-D = CMAC D blade; CMAC Miller = CMAC Miller blade; CMAC-PM = CMAC pocket monitor; Dahlhausen = Dahlhausen; DCI = Storz-DCI; GS = GlideScope; GS-AVL = GlideScope Advanced Video Laryngoscope; GS Cobalt = GlideScope Cobalt; Imago-V-C = Imago channelled V blade; IB = Intubrite; KV = King Vision; KV-C = King Vision channelled; KV-NC = King Vision non-channelled; KV-aBlade-C = King Vision aBlade channelled; KV-aBlade-NC = King Vision aBlade non-channelled; KV paeds aBlade = King Vision Paediatric aBlade; McGrath Mac = McGrath Mac; McGrath Mac (acute) = McGrath Mac acute-angled blade; McGrath Series 5 (acute) = McGrath Series 5 acute-angled blade; McGrath Mac-X = McGrath Mac X-blade; O-Mac = O-Mac; ST = Smart Trach; TV-PCD = Truview PCD; TV = Truview; TV EVO_2_ = Truview EVO_2_; Wycope = Wycope.

**Table 3 healthcare-11-02383-t003:** Outcomes of comparative studies in simulation studies using different types of videolaryngoscopes.

Year	First Author	No. of Providers	Simulation Setting	Intubator	Number and Types of Videolaryngoscopes Used	Outcomes Based On: A: Improved Glottis View B: Faster Time to Successful Intubation C: Higher First Attempt Intubation Success D: Less Use of Manoeuvres or Adjuncts E: Other	Clinically Preferred Videolaryngoscope	Blade of Most-Preferred VL
2016	Altun [58]	41	Difficult Manikin	Anaesthetic Residents	2: McGrath Mac (acute), CMAC	A: CMAC B: CMAC C: CMAC D: CMAC	CMAC	Non-channelled, Macintosh
2016	Arslan [59]	36	Difficult Paediatric Manikin	Medical Students	2: GS, AT	A: - B: - C: GS D: GS	GS	Non-channelled, Acute-angled
2016	El-Tahan [60]	21	Double Lumen Tube Manikin	Anaesthetists	3: GS, AT, KV-NC	A: Equal B: Equal C: Equal D: GS > KV-NC E: GS Preferred over AT and KV-NC	GS	Non-channelled, Acute-angled
2016	Hippard [61]	30	Paediatric Manikin	Anaesthetists	2: TV-PCD, GS-Cobalt	A: - B: Equal C: Equal D: -	TV-PCD = GS-Cobalt	TV-PCD = Non-channelled, Acute-angled GS-Cobalt = Non-channelled, Acute-angled
2016	Kim [62]	35	Manikin	Physicians	3: AWS, GS	A: GS B: Equal C: GS D: -	GS	Non-channelled, Acute-angled
2016	Kim [63]	21	Manikin	Physicians	2: AWS, GS	A: Equal B: AWS C: Equal D: -	AWS	Channelled
2016	Nakanishi [64]	35	Manikin	Physicians	2: AWS, CMAC	A: Equal B: AWS C: Equal D: - E: Higher Force on Incisors with CMAC than AWS	AWS	Channelled
2016	Schröder [65]	42	Manikin wearing chemical protective gear	Anaesthetists	3: AT, GS, AP Advance	A: AP Advance > AT and GS B: AP Advance > AT and GS C: - D: - E: Preferred AP Advance > GS > AT	AP Advance	Channelled
2016	Shin [66]	39	Manikin	Novice Medical Students	2: McGrath Mac, CMAC	A: Equal B: Equal C: Equal D: Equal E: Preferred McGrath Mac	McGrath Mac	Non-channelled, Macintosh
2017	Hodnick [67]	5	Cadaver	Paramedics	2: GS, VT	A: Equal B: Equal C: Equal D: Equal	GS = VT	GS = Non-channelled, Acute-angled VT = Channelled
2017	Lee [68]	18	Manikin with Normal and Difficult Airway	Paramedics	4: GS, AWS, KV-NC, KV-C	A: Equal B: AWS C: Equal D: - E: Preferred AWS	AWS	Channelled
2017	Owada [69]	20	Paediatric Manikin with Difficult Airway	Anaesthetists	2: AT, McGrath Mac	A: AT B: Equal C: AT D: - E: AT Preferred and less dental trauma	AT	Channelled
2018	Kriege [70]	80	Infant Manikin Normal and Difficult Airway	Anaesthetic Staff and Paediatric Critical Care Medicine Staff	2: KV-Paeds aBlade, CMAC-D	A: KV-Paeds aBlade B: KV-Paeds aBlade C: KV-Paeds aBlade D: -	KV-Paeds aBlade	Non-Channelled, Acute-angled
2018	Oshika [71]	21	Prone Manikin	Anaesthetists	2: AWS, McGrath Mac	A: - B: AWS C: AWS D: -	AWS	Channelled
2019	Chew [72]	105	Difficult Airway Manikin	Junior Doctors	3: KV-aBlade-C, KV-aBlade-NC, Mcgrath Mac	A: - B: KV-aBlade-C + McGrath Mac > KV-aBlade-NC C: KV-aBlade-C + McGrath Mac > KV-aBlade-NC D: -	KV-aBlade-C + McGrath Mac	KV-aBlade-C = Channelled McGrath Mac = Non-channelled, Macintosh
2019	Desai [73]	26	Paediatric Pierre Robin Sequence Manikin	Paediatric Intensive Care Physicians	2: AT, GS	A: Equal B: AT C: AT D: -	AT	Channelled
2019	Raimann [74]	42	Trapped Car Crash Manikin	Anaesthetic Staff and Emergency Physicians	4: CMAC-D, TV-PCD, CMAC, CMAC-PM	A: CMAC-D > TV-PCD B: CMAC > TV-PCD + CMAC-D, CMAC-PM > TV-PCD C: CMAC + CMAC-PM > TV-PCD D: - E: CMAC Preferred	CMAC	Non-channelled, Macintosh
2020	Gaszyński [75]	11	Prone and Sitting Manikin	Anaesthetists	2: AWS, IB	A: Equal B: AWS C: Equal D: - E: Lower pressure with AWS	AWS	Channelled
2020	Moritz [76]	112	Difficult Airway Manikin	Anaesthetist and Paramedics	4: I-View, KV-NC, GS, Dahlhausen	A: KV-NC, GS, Dahlhausen > I-View B: KV-NC, GS, Dahlhausen > I-View C: KV-NC, GS, Dahlhausen > I-View D: - E: Preferred GS	GS	Non-channelled, Acute-angled
2020	Romito [77]	8	Cadaver with Cervical Spine Instability	Anaesthetists	3: CMAC-D, GS, McGrath Mac-X	A: Equal B: - C: Equal D: -	CMAC-D = GS = McGrath Mac-X	CMAC-D = Non-channelled, Acute-angled GS = Non-channelled, Acute-angled McGrath Mac-X = Non-channelled, Acute-angled
2020	Votruba [78]	58	Manikin with Cervical Spine Immbolisation	Anaesthetists	2: KV-C, KV-NC	A: Equal B: KV-C C: KV-C D:- E: Easier with KV-C	KV-C	Channelled
2020	Yi [79]	35	Manikin with Normal Neck and with Cervical Spine Instability	Anaesthetic Nurses	2: McGrath Mac, AWS	A: Equal B: Equal C: Equal D: -	McGrath Mac = AWS	MGrath Mac = Non-channelled, Macintosh AWS = Channelled
2021	Ataman [80]	23	Manikin with Normal and Difficult Airway	Emergency Physicians and Emergency Residents	2: GS, AirAngel 3D-Printed	A: - B: GS C: GS D: -	GS	Non-channelled, Acute-angled
2021	Decamps [81]	79	Critical Illness Manikin	Residents	4: KV-C, AWS, AT, VT	A: KV-C + AWS > VT B: AWS > VT C: Equal D: Equal E: Ease of use with KV-C + AWS + AT > VT	AWS	Channelled
2021	Gupta [82]	50	COVID Simulation Manikin	Anaesthetist and Non-Anaesthetic Physicians	2: KV-C, Tuoren	A: KV-C B: KV-C C: KV-C D: KV-C E: Easier and less complications with KV-C	KV-C	Channelled
2021	Moritz [83]	86	Paediatric Pierre Robin Sequence Manikin	Anaesthetists	2: GS-Core, CMAC-Miller	A: GS B: CMAC-Miller C: Equal D: CMAC-Miller E: CMAC-Miller preferred by both anaesthetist with experience and with limited experience	CMAC-Miller	Non-channelled, Miller
2021	Taylor [84]	33	Manikin	Military emergency providers	2: I-view, GS	A: GS > I-view B: GS > I-view C: Equal D: -	GS	Non-channelled, Acute-angled
2021	Vig [85]	30	Manikin	Medical Professionals (no intubation experience)	2: McGrath Mac, CMAC	A: CMAC > McGrath B: Equal C: Equal D: Equal	CMAC	Non-channelled, Macintosh
2022	Gupta [86]	100	Manikin	Medical Students	2: KV-C, KV-NC	A: Equal B: KV-C > KVNC C: KV-C > KVNC D:- E: KV-C > KVNC ease of intubation	KV-C	Channelled
2022	Er [87]	50	Manikin	Paediatric Residents	2: McGrath Mac, CMAC PM	A: CMAC PM > McGrath Mac B: CMAC PM > McGrath Mac C: CMAC PM > McGrath D:-	CMAC	Non-channelled, Macintosh
2022	Kumar [88]	61	Manikin	Healthcare Staff with COVID PPE	2: CMAC, KV-C	A: Equal B: Equal C: Equal D: Equal E: CMAC > KV-C—easier insertion of laryngoscope blade	CMAC	Non-channelled, Macintosh

AirAngel 3D-printed = AirAngel blade 3D-printed; APA = Venner AP advance; AWS = Pentax Airway Scope; AT = Airtraq; CMAC = CMAC; CMAC-D = CMAC D blade; CMAC Miller = CMAC Miller blade; CMAC-PM = CMAC pocket monitor; GS = GlideScope; GS Cobalt = GlideScope Cobalt; GS-Core = GlideScope Core; I-view = I-view; KV-aBlade-C = King Vision aBlade channelled; KV-aBlade-NC = King Vision aBlade non-channelled; KV-C = King Vision channelled; KV-NC = King Vision non-channelled; KV paeds aBlade = King Vision Paediatric aBlade; McGrath Mac = McGrath Mac; McGrath Mac (acute) = McGrath Mac acute-angled blade; Tuoren = TUORen; TV-PCD = Truview PCD; VT = VividTrac VT-A100.

**Table 4 healthcare-11-02383-t004:** Outcomes of videolaryngoscopy studies listed by most-preferred VLs.

	Surgical				Simulation				Total		
VLS	Surgical Studies	Favoured Studies	% of Studies Favoured	Total Number of Patients in Studies	Simulation Studies	Favoured Studies	% of Studies Favoured	Total Number of Participants in Studies	Total Studies	Favoured	% Favoured
CMAC	25	19	76%	2894	12	7	58%	472	37	26	70%
McGrath	19	12	63%	2304	9	4	44%	349	28	16	57%
GS	13	9	69%	1240	14	9	64%	496	27	18	67%
KV-NC	6	5	83%	1074	7	1	14%	494	13	6	46%
AWS	8	4	50%	1001	8	7	88%	255	16	11	69%
AT	11	6	54%	1397	6	2	33%	224	17	8	47%
KV-C	12	2	17%	1756	7	4	57%	471	19	6	32%

CMAC = CMAC; CMAC D-blade; CMAC Miller blade; CMAC Pocket Monitor; McGrath = McGrath Mac; McGrath Series 5; McGrath Mac X-blade; GS = GlideScope; GlideScope Cobalt; GlideScope Advanced Video Laryngoscope; KV-NC = King Vision non-channelled; King Vision aBlade non-channelled; King Vision Paediatric aBlade; AWS = Pentax Airway Scope; AT = Airtraq; Airtraq double lumen; KV-C = King Vision channelled; King Vision aBlade channelled.

## Data Availability

Data is contained within the article.

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
