# Peer review of "Clinically Preferred Videolaryngoscopes in Airway Management: An Updated Systematic Review"

_healthcare, 2023, doi:10.3390/healthcare11172383_

Round 1

Reviewer 1 Report

The manuscript was well written with good language and layout. And your conclusions are good reference for the young physicians in their future clinical practice.

Line 220 ...such 'a' preparing a bougie--> such 'as' ?

Some other typos have to be corrected.

Author Response

Thank you kindly for your support of this article and valuable feedback. We have reviewed the article and corrected our identified typos

  • Line 220: changed ‘a’ to ‘as’

Reviewer 2 Report

This is a very good review of an important topic that has not received much attention to date. The methodology is sound and the presentation interesting.

There are a few grammatical and spelling errors that should be attended to in the final edit.

There are a few grammatical and spelling errors that need to be attended to in the finale edit.

Author Response

Thank you kindly for your support of this article and valuable feedback. We have reviewed the article and corrected our identified grammatical and spelling errors.

Reviewer 3 Report

Thorough review of current literature regarding video laryngoscopy.  Can. not say though that CMAC, GlideScope or Pentax are clinically preferred without a survey of users.  Can say that there is more published work using CMAC, GS or Pentax but it is not know if these publications reflect actual use.

The other factor missing in this review is what type of patients were identified for video laryngoscopy?  Were first pass attempts compared to DL in a randomized manner or were certain patients identified as needing video laryngoscopy.  Was VL used primarily in teaching centers or in community hospitals?

Fundamentally can not state which VL system is most preferred based upon what has been published.  Good points made about minimal head to head comparisons and differences between manikin and patient studies.

Need to better distinguish when different VL systems might be used.  In a teaching environment, CMAC might be strongly preferred for the reasons outlined in manuscript.  However, for difficult airway intervention, Glidescope or similar is often used.  Hence a broad review of the literature is not going to state when one system is preferred over another.

Author Response

Thank you kindly for your support of this article and valuable feedback.

We certainly acknowledge that preference cannot be determined without a direct survey. We have made addendums to suggest the data may represent more commonly used VLs, and future studies evaluating more head-to-head comparisons may lead to more direct comparisons and thus more direct preferences.

  • Addition, Line 197-200 (new): We also propose that the current literature favours the most commonly used VLs, and true clinician preference would be better ascertained in future studies evaluating more head-to-head comparisons and thus more direct parameters for clinician preference.

We agree certain VLs may be more preferred in particular settings and thank you for highlighting this. We have made addendums to ensure it is clear we agree and believe future studies should identify patient factors and institutional factors that may lead to particular VLs being more preferred. We hope this article lays a platform to highlight the need for future studies on what we believe is an important airway topic.

  • Addition, Line 233-236 (new): We also advocate for future studies to more thoroughly identify patient and institutional factors that may lead to the use of, or preference for, a particular VL, with the aim to allow transparency to readers regarding scenarios where particular VLs may be more favourable.

Reviewer 4 Report

Thank you for permitting me to review this manuscript 

A picture of  most preferred  models or available would have been beneficial to better drive the reader 

It should be noted that in most hospitals , there might only be one or two types of VL , therefore it is difficult to get used with all models available ,  thereforz studies are necessary to guide the choice of a departement , not individual anesthesia providers 

A mini summary of  the 2016 work will better  help the reader in the introduction instead of current discussion  , 

Line 262 to 264 is unnecessary 

Author Response

Thank you kindly for your support of this article and valuable feedback. We have added images of the most preferred VLs to better engage the reader.

You highlight a key point regarding the availability of VLs in hospitals. We agree and feel this is an important aspect of clinician/department preference for VLs which future studies should evaluate as highlighted and since expanded on, thanks to your suggestion. 

  • Addition, Line 233-236 (new): We also advocate for future studies to more thoroughly identify patient and institutional factors that may lead to the use of, or preference for, a particular VL, with the aim to allow transparency to readers regarding scenarios where particular VLs may be more favourable.

Our previous study mini summary is now highlighted in the introduction as well, thank you.  

  • Addition, Line 65: Our previous study published in 2016 identified…

We have edited Lines 262 to 264 to be more fitting with this article

  • Addition, Line 261-263: We hope this audit increases awareness to both individual practitioners, but also to departments of Anaesthesia to highlight the importance of VL use and to lay a platform for future studies to expand knowledge in this field

Reviewer 5 Report

Regarding the manuscript entitled: "Clinically preferred videolaryngoscopes in airway management. An updated systematic review", I would like to congratulate the authors. This review provides evidence on the characteristics of the different videolaryngoscopes and the alternatives of the anesthesiologists in the approach to the airway.

Introduction:

The authors dedicate an extensive paragraph to the preoperative evaluation of the airway (lines 37-45). From my point of view, this information can be eliminated/summarized, since it is not related to the subject of study.

The information provided between lines 55-65 could be summarized so that it is not so extensive to the potential reader.

In addition, I consider that the information provided in lines 66 - 76 is better to include in the Discussion section since it can be extracted from the information provided by the results.

Materials and Methods:

Did the authors register the systematic review in PROSPERO? Despite not being mandatory, it is highly recommended to carry out a prospective registry to increase quality and avoid bias.

Did the authors follow the PRISMA checklist when writing the article? I recommend that this section be structured into different sections to increase the possibility of replication of the systematic review and to better understand the methodology followed: Eligibility criteria; search strategy; study selection; Data extraction.

It should be noted in the text where figure 1 and table 1 are identified.

However, I consider that Table 1 can be omitted (I don't see any sense in the description of the different videolaryngoscopes in this section).

Results:

Tables 2 and 3 are excessively long and difficult to understand. I consider that they should be changed, summarized or directly eliminated from the text. They could be added as supplementary material. It is ineffective for the article to expose so much information that has not previously been brought together.

I think the results section should be restructured to provide relevant information from the many published studies.

Discussion:

Much of the information provided in the conclusions section can be included in the Discussion section (purpose of the study, comparison with previous reviews, potential future studies, what can be improved to have optimal results, etc.).

Conclusions:

In my opinion, the conclusions section should be limited to extracting the most important data from the study.

Author Response

Thank you kindly for your support of this article and valuable feedback.

We appreciate you highlighting the edits as suggested. We have rationalised and summarised the suggestions in the Introduction. With regards to your suggestion about Lines 55-65, we found it difficult to summarise this any further without negatively impacting the key points, but do definitely acknowledge your feedback.

  • Line 37-40 (old) REMOVED: A typical pre-operative anaesthetic assessment includes airway assessment (mouth opening, thyromental distance, Mallampati class, ability to prognath the lower jaw, lip bite test), evaluation of neck (circumference and range of motion in flexion-extension) and information about the patient's weight, body mass index (BMI) and history of previous difficulties with intubations
  • Line 66-72 (old) REMOVED and reworded in Discussion: Commonly, VLs are divided into two categories - channelled or non-channelled. Channelled VLs, such as the Pentax-AWS or Airtraq, have an endotracheal tube loaded within a channel attached to the blade ready to be inserted under vision. On the other hand, non-channelled VLs, such as McGrath and CMAC do not have a channel and the tracheal tube must be guided manually into the trachea under vision and inserted separately, potentially with the aid of adjuncts

The study was registered in PROSPERO. We have now mentioned this in the Materials and Methods for transparency. Thank you for your suggestion regarding PRISMA and structure. We did follow PRISMA checklist and have restructured the methods to follow the structure you recommend.  

  • Addition, Line: 81-83: A PRISMA checklist was used to ensure optimal strategy with a summary provided in Figure 1. This study was registered in PROSPERO.

We acknowledge your comment regarding Table 1. We believe this table adds value to a comparison to our previous study in 2016, as well as an easy visualization for readers to demonstrate the current VLs which have been studied and available. Similarly, Table 2 and 3 are used in addition to our previous study in 2016. We agree, this may be more beneficial as supplementary material, if the journal is willing to consider this as an option.

With regards to the discussion and conclusion, we have restructured our format as you kindly suggested to be more succinct and streamlined. We hope this covers your key areas and fulfills the requirements for publication.

  • Line 246-252 (moved from Conclusion to Discussion with minor edits): To achieve optimal clinical outcomes, we advocate that clinicians choose the VL that they are most comfortable with depending on the clinical situation, as each VL offers its own set of advantages and disadvantages. We believe that VL should become the gold standard for ETI, with several advantages highlighted over DL. Furthermore, VL can also allow us to transition from blind insertion techniques for procedures including temperature probes and nasogastric tubes to ‘vision-guided’ insertion, ultimately limiting the potential for wrong space insertion, with future studies potentially exploring this field.

Round 2

Reviewer 3 Report

Thank you for addressing the prior comments and revising your manuscript.  No further concerns

Reviewer 5 Report

The authors have responded to most of the suggestions made.